# Development of a Molecular Snail Xenomonitoring Assay to Detect *Schistosoma haematobium* and *Schistosoma bovis* Infections in their *Bulinus* Snail Hosts

**DOI:** 10.3390/molecules25174011

**Published:** 2020-09-02

**Authors:** Tom Pennance, John Archer, Elena Birgitta Lugli, Penny Rostron, Felix Llanwarne, Said Mohammed Ali, Amour Khamis Amour, Khamis Rashid Suleiman, Sarah Li, David Rollinson, Jo Cable, Stefanie Knopp, Fiona Allan, Shaali Makame Ame, Bonnie Lee Webster

**Affiliations:** 1Wolfson Wellcome Biomedical Laboratories, Department of Life Sciences, Natural History Museum, Cromwell Road, London SW7 5BD, UK; e.lugli@nhm.ac.uk (E.B.L.); p.rostron@nhm.ac.uk (P.R.); felix.llanwarne1@student.lshtm.ac.uk (F.L.); d.rollinson@nhm.ac.uk (D.R.); f.allan@nhm.ac.uk (F.A.); b.webster@nhm.ac.uk (B.L.W.); 2School of Biosciences, Cardiff University, Cardiff CF10 3AX, UK; cablej@cardiff.ac.uk; 3London Centre for Neglected Tropical Disease Research (LCNTDR), London W2 1PG, UK; 4Faculty of Infectious and Tropical Diseases, London School of Hygiene and Tropical Medicine, Keppel Street, London WC1E 7HT, UK; 5Public Health Laboratory–Ivo de Carneri, P.O. Box 122 Chake-Chake, Pemba, Tanzania; saidmali2003@yahoo.com (S.M.A.); amourkhamis2003@yahoo.com (A.K.A.); khasule66@yahoo.com (K.R.S.); shaaliame@yahoo.com (S.M.A.); 6Schistosomiasis Resource Centre, Biomedical Research Institute, 9410 Key West, Rockville, MD 20850, USA; sli@afbr-bri.org; 7Swiss Tropical and Public Health Institute, Socinstrasse 57, 4002 Basel, Switzerland; s.knopp@swisstph.ch; 8University of Basel, Petersplatz 1, 4001 Basel, Switzerland

**Keywords:** bovine, control, elimination, schistosomiasis, urogenital, surveillance, disease, parasite

## Abstract

Schistosomiasis, a neglected tropical disease of medical and veterinary importance, transmitted through specific freshwater snail intermediate hosts, is targeted for elimination in several endemic regions in sub-Saharan Africa. Multi-disciplinary methods are required for both human and environmental diagnostics to certify schistosomiasis elimination when eventually reached. Molecular xenomonitoring protocols, a DNA-based detection method for screening disease vectors, have been developed and trialed for parasites transmitted by hematophagous insects, such as filarial worms and trypanosomes, yet few have been extensively trialed or proven reliable for the intermediate host snails transmitting schistosomes. Here, previously published universal and *Schistosoma-*specific internal transcribed spacer (ITS) rDNA primers were adapted into a triplex PCR primer assay that allowed for simple, robust, and rapid detection of *Schistosoma haematobium* and *Schistosoma bovis* in *Bulinus* snails. We showed this two-step protocol could sensitively detect DNA of a single larval schistosome from experimentally infected snails and demonstrate its functionality for detecting *S. haematobium* infections in wild-caught snails from Zanzibar. Such surveillance tools are a necessity for succeeding in and certifying the 2030 control and elimination goals set by the World Health Organization.

## 1. Introduction

Schistosomiasis is a disease affecting an estimated 229 million people worldwide caused by infection with parasitic worms of the genus *Schistosoma*, leading to severe morbidity and mortality due to the associated complications of worm presence [1]. *Schistosoma* spp. in Africa are transmitted through specific freshwater snail intermediate hosts of the *Bulinus* and *Biomphalaria* genera [2]. Infections occur when humans or animals come into contact with freshwater containing infectious larval stages (cercariae) shed from the infected snails. Human schistosomiasis in Africa, where at least ~90% of the people requiring treatment live [3], consists of two forms of the disease—urogenital and intestinal schistosomiasis, caused predominantly by *Schistosoma haematobium* and *Schistosoma mansoni,* respectively [1]. Bovine, ovine, and caprine schistosomiasis is also of significant veterinary and economic importance across sub-Saharan Africa [4,5] and is caused by infection of cattle, sheep, and goats with species closely related to *S. haematobium* (termed *S. haematobium* group species), primarily *Schistosoma bovis, Schistosoma curassoni,* and *Schistosoma mattheei*. Overlapping geographical distribution of multiple schistosome and intermediate snail host species strains complicates disease transmission surveillance in (co)endemic zones [2,6,7].

The World Health Organization (WHO) aims for the elimination of human schistosomiasis as a public health problem, defined as >1% of the population with heavy intensity infections (≥50 schistosome eggs per 10 mL of urine, or ≥400 schistosome eggs per gram of feces [8]), in all endemic countries by 2030 [9]. Despite great advances in schistosomiasis control mainly via preventative chemotherapy (praziquantel), the lack of protection against rapid re-infection together with prolific asexual replication of schistosomes within their intermediate snail host presents substantial hurdles to achieving the targeted elimination of schistosomiasis. Very quickly, snails can become infected by eggs emanating from untreated humans, leading to a rapid resurgence of transmission [10]. Therefore, adaptive treatment strategies that take into account the transmission dynamics of *Schistosoma* spp. with their snail hosts are required to control and eliminate the disease [11].

To better understand the local transmission dynamics of different *Schistosoma* species, allowing both human and bovine schistosomiasis to be monitored, a need exists for methodologies that detect schistosome infections in the intermediate host snails. These tools for assessing *Schistosoma* transmission could eventually be used during elimination programs to identify focal areas of persisting transmission or certify elimination and/or transmission interruption [12,13,14]. Defining ongoing transmission in snail populations through traditional methods of observing cercariae shed from snails is particularly challenging in an elimination setting, such as the Zanzibar Archipelago, where few snails (0.5–2.3%) are observed shedding cercariae [6,15]. Snails with non-patent (including pre-patent) infections are missed using these approaches. Additionally, larval schistosomes are not easily identifiable to a species level using morphological characteristics (although the relative position of sensory receptors is of some value [16,17]).

Molecular xenomonitoring is a DNA-based method that has been developed to monitor the transmission of several vector-borne diseases, including trypanosomiasis [18,19], filariasis and malaria [20], helminthiases [21], and fascioliasis [22], including to some extent schistosomiasis [23,24,25,26,27,28,29]. Screening snails provides evidence on the extent of environmental contamination (i.e., schistosome miracidia penetrating snails), as well as environmental infection risk (i.e., schistosome sporocysts and cercariae developing inside the (pre-patent) snails, eventually emerging from the (patent) snail. Most of the available snail-schistosome xenomonitoring assays do not include internal controls [23,28,30], an important feature in any diagnostic tool that helps prevent false-negative results [27]. Many assays will assume that a negative result means non-infection, not necessarily reaction failure.

In the current study, we adapted available universal [31] and *Schistosoma-*specific [27] internal transcribed spacer (ITS) rDNA primers to design a three primer multiplex assay and tested this as a simple, robust, and rapid xenomonitoring PCR assay to enable the large-scale screening of *Bulinus* snails for *Schistosoma* infections (*S. haematobium* and *S. bovis*). We used a conventional PCR-based approach focused on simplicity, ease of data interpretation, sensitivity, and specificity, with a primary aim to provide a xenomonitoring tool for monitoring *S. haematobium* transmission in endemic settings.

## 2. Results

### 2.1. In Silico and In Vitro Primer Evaluation

*Bulinus globosus* and *Bulinus nasutus* rDNA sequence data showed conserved primer binding sites for the universal primers ETTS2 and ETTS1 [31] at the 3′ end of the 18S and 5′ end of the 28S, the flanking regions of the ITS, respectively. ETTS1 gave a 100% match, and the ETTS2 primer showed just a single base pair mismatch. The resulting snail amplicon size predicted from these alignments was between 1232 and 1263 bp and served as an internal snail control during PCR amplification.

Alignments of the *Schistosoma-*specific ITS primers (ITS2_Schisto_F and ITS2_Schisto_R [27]) showed 100% and 90% (2 mismatches) homology to *S. haematobium* and *S. bovis,* respectively, with no cross-reactivity to the *Bulinus* reference rDNA data. When paired with their opposing universal primers (ITS2_Schisto_F + ETTS1 or ITS2_Schisto_R + ETTS2), the amplicon sizes of 538 and 835 bp were predicted, respectively, for *Schistosoma*. With the addition of the other universal primer to each combination (ETTS2 and ETTS1, respectively), the three-primer multiplex ITS xenomonitoring (MIX) reactions were predicted to be able to produce distinct amplicon profiles for non-infected snails (a single snail amplicon) and snails infected with *Schistosoma* spp. (three-band profile). This was confirmed by in vitro testing of the primer combinations (Figure 1).

To maximize amplification efficiency/sensitivity and to provide good amplicon size differentiation, the multiplex PCR incorporating the internal ITS2_Schisto_F (Figure 1B) was selected for further development and testing. This primer combination was also selected as it targeted the ITS2 region for *Schistosoma* containing four species-specific SNPs, enabling species identification (Table 1).

The MIX assay proved robust at varying annealing temperatures (55 °C, 60 °C Figure 2A, 58 °C Figure 2B), with 58 °C proving to be the most efficient, maximizing specificity without decreasing sensitivity. Each of the three amplicons was extracted from the gel and sequenced, confirming the band identity and specificity of the primers to their target gDNA amplicon. These three bands have been described as the snail (Sn) (1232–1263 bp), trematode (T) (~1000 bp), and *Schistosoma* (S) (538 bp) bands going forward. The secondary *Schistosoma* ITS xenomonitoring (SIX) PCR, solely targeting the *Schistosoma* amplicon, proved robust, enabling single amplicon generation and sequencing (Figure 3). This provided a two-step PCR methodology with the MIX PCR for the initial high-throughput screening of the samples and the secondary SIX PCR to target specific samples for further infection clarification by *Schistosoma* species identification through DNA sequencing.

### 2.2. Analytical Sensitivity

The assay proved highly sensitive with a limit-of-detection (LoD) of 0.02 ng and 0.002 ng of gDNA for *S. bovis* and *S. haematobium*, respectively (Figure 4). Sensitivity appeared higher for *S. haematobium* (Figure 4), but in both the cases, the assay’s sensitivity was above that necessary to detect gDNA from a single miracidium, which ranges from 1.6–3.65 ng/µL [32]. At lower *Schistosoma* DNA concentrations, the 1005 bp trematode band (T) lost sensitivity compared with the smaller *Schistosoma-*specific band. 

### 2.3. Experimental Snail Infections

For the non-patent infections of *Bulinus truncatus* with *S. haematobium,* preserved 24 h after exposure, 61.1% (11 out of 18) of the snails were observed to be penetrated by the *S. haematobium* miracidia, presenting the *Schistosoma-*specific ITS2 band (Figure 5). Infections were detected in snails exposed to 1, 2, and 7 miracidia. Two of the five (40%) *B. truncatus,* exposed to one or two miracidia and left for 11 weeks, did not reach patency but were also confirmed to be penetrated by *S. haematobium* miracidia (Figure 5: Lanes 20 and 21). The secondary SIX PCR was performed on all 13 non-patent infected snails, and the single amplicons were sequenced and confirmed as *S. haematobium*. Out of all the snails infected that survived until the end of the experiment (11 weeks), 15% (nine out of 62) reached patency, of which two had been infected with two miracidia, and seven with seven miracidia. One of these samples, infected with two miracidia, was analyzed using the MIX PCR, giving the expected triple banding pattern (snail, trematode, and *Schistosoma*) (Figure 5: Lane 24). All three amplicons from this sample (Figure 5: Lane 24) were gel extracted and sequenced, confirming their identity. Interestingly, in all the non-patent infections, the large trematode amplicons (ETTS2-ETTS2) did not amplify (Figure 5) due to the low level of *Schistosoma* DNA present in the snails that did not reach patency.

### 2.4. Specificity Testing

The *B. globosus* with patent *S. haematobium* (*n* = 2) and *S. bovis* (*n* = 5) infections showed the expected triple banding pattern (snail, trematode, and *Schistosoma* amplicons, results not shown), and following gel extraction and sequencing the data matched those from the cercariae collected from these samples (GenBank Accessions: MH014047 and MH014044, see [6]). 

When the MIX PCR was tested on snails confirmed to be infected with other commonly found trematode species (*B. globosus* infected with *Euclinostomum* sp., and the *B. nasutus* infected with *Echinostoma* sp. (Figure 5: Lanes *Euc.* and *Ech.*)), no *Schistosoma* amplicon was observed. However, there was strong amplification of the trematode band together with the snail band. These trematode amplicons were gel extracted, sequenced, and the infections were confirmed as *Euclinostomum* sp. and *Echinostoma* sp., matching data from the cercariae originally collected from each snail.

### 2.5. Testing on Field Samples 

From the 94 field-collected *B. globosus,* 33 were shown to be infected with *Schistosoma* spp. with amplification of the *Schistosoma-*specific band (Figure 6). Among them, eight also presented the trematode band. The internal snail control was amplified in all samples apart from one. The one that failed was predicted to be due to poor sample preservation, gDNA extraction, or PCR error, and so was disregarded (Figure 6). Of all the samples that gave the *Schistosoma-*specific band, the secondary SIX PCR (ITS2_Schisto_F + ETTS2) was conducted, and all amplicons were Sanger sequenced. Two failed to amplify, but the remaining 31 produced the *Schistosoma* amplicon that all sequenced as *S. haematobium*. One sample also gave the trematode band without the *Schistosoma* band, indicating a non-*Schistosoma* trematode infection.

### 2.6. Schistosoma spp. cox1 RD-PCR

Despite trying different annealing temperatures and gDNA template amounts used, the *cox*1 RD-PCR, developed by Webster et al. (2010) [33], tested on the patent *S. haematobium* and *S. bovis-*infected *B. globosus,* only generated the species-specific amplicon for *S. bovis-*infected snails. PCRs for the *S. haematobium-*infected snails repeatedly failed to produce a clear amplicon. The *cox*1 amplicons produced for the *S. bovis-*infected snails were sequenced, and the data matched that obtained from the cercariae collected and analyzed from the snails (see [6]).

## 3. Discussion

Pre-patent and non-patent snail screening methods for schistosomes, such as molecular xenomonitoring, offer a higher sensitivity over traditional snail shedding methods that can only detect patent infections by the observation of schistosome cercariae. Molecular xenomonitoring better helps to assess the impact of schistosomiasis control interventions in local communities, particularly where local elimination is being achieved, and certification of the absence of transmission is required at specific foci. However, the diversity of schistosomes circulating in co-endemic areas means that species-specific methods are needed to prevent false-positive data due to non-target species cross-reactivity.

Here, we describe the development and application of a molecular xenomonitoring pipeline for the detection and differentiation of *S. haematobium* and *S. bovis* patent and non-patent infections in *Bulinus* freshwater snails, using three previously developed primers [27,31]. The MIX assay screens for *Schistosoma* and other trematode species, while also incorporating an internal control, in this case, gastropod DNA, an important feature for any molecular diagnostic assay. The MIX PCR generates clearly identifiable amplicons, of different sizes, for each target (snail, trematode, *Schistosoma*), which are visible by simple agarose gel electrophoresis. However, the trematode target lacks sensitivity at low DNA concentrations, probably due to its large size and PCR biases for small amplicons at reduced gDNA concentrations. Interestingly, a PCR artifact (~1400–1600 bp) is also observed when using the MIX assay in the presence of *Schistosoma* DNA, suggesting that the primers may have a secondary binding site. However, this artifact is clearly identifiable from the main target amplicons and does not mislead the interpretation of the results.

### 3.1. The Sensitivity of MIX PCR Assay

Our in silico and in vitro testing of the MIX assay shows that the presence of *S. haematobium* and *S. bovis* DNA can be routinely detected at low concentrations and also is able to identify non-patent *Schistosoma* infections in snails where the level of DNA varies depending on the development of the infection. The LoD for *Schistosoma* DNA is ≤0.02 ng/µL, which is 80-fold lower than the minimum amount of gDNA usually observed from a single miracidium [32]. This is also demonstrated by the assay’s ability to detect pre-patent snail infections 24 h after exposure to a single miracidium. This provides sufficient sensitivity for the LoD needed to detect any stage of snail infection, from initial miracidia penetration of a single miracidium to full patency, in natural settings. The fact that not all the snails tested from the experimental snail infections give positive results is corroborative with observations that, even in the experimental systems, many snails avoid penetration or destroy the miracidia rapidly upon invasion. The MIX and SIX methodologies also prove robust when used to screen ‘wild-caught’ snails from Pemba, with uninfected, pre-patent *S. haematobium-*infected snails, and non-*Schistosoma* trematode infections are clearly identified.

### 3.2. Benefits of an Updated Molecular Xenomonitoring Protocol for Schistosomiasis Surveillance

The molecular xenomonitoring protocol requires few consumables and no cold chain, and results can be interpreted using basic molecular laboratory equipment (thermocycler and gel electrophoresis), making the molecular assay accessible in lower resource settings, such as schistosomiasis endemic regions. The molecular xenomonitoring approach described here, therefore, provides a useful tool for monitoring schistosomiasis transmission, as has been outlined as a necessary method for leading toward the WHO 2030 goals for schistosomiasis control and elimination [9].

Molecular xenomonitoring surveillance techniques are often associated with parasites transmitted by hematophagous insects, such as lymphatic filaria in mosquito vectors [20,34,35,36] and trypanosomes in tsetse flies [18,19]. However, several assays have been developed for detecting trematode species in freshwater snails, including *Fasciola* spp. [22,37,38,39,40,41,42,43,44], other wildlife trematode species [45], and medically important schistosome species—*S. japonicum* [46,47], *S. mansoni* [24,27,28,48,49,50,51,52,53,54]*,* and *S. haematobium* [23,26,27,30,52,54,55]. The first developed assay for the molecular detection of *S. haematobium* DNA in *Bulinus* employs the highly repetitive *Dra*1 target, and this has been the marker of choice for studies investigating *S. haematobium* infections in snails due to its high sensitivity [55]. However, the specificity of the *Dra*1 and the interpretation of results can be problematic due to the frequent false-positive and -negative results, lack of internal control, and difficulties in interpreting the amplicon patterns. Furthermore, this marker does not allow for species identification. Kane et al. (2013) [54] employed the use of another repetitive marker, intergenic spacer (IGS), for the detection of snail infections, and a post-amplification restriction digest allowed for the downstream species identification of *S. haematobium* and *S. bovis*. However, the method lacks internal controls. In addition, many of these assays use quantitative-PCR (qPCR), rather than conventional PCR/gel electrophoresis. Although able to quantify levels of DNA within a sample, qPCR is more arduous to carry out and lesser suited for use in endemic settings. However, recent technological advances in sample preparation and DNA extraction methods have demonstrated robust field setting methodologies to conduct qPCR analysis capable of detecting avian trematodes and host species in Canadian lakes [56,57,58], which could potentially be modified to suit the detection of human and bovine schistosomes in sub-Saharan Africa, although cost and throughput would need to be considered.

A recent assay designed by Schols et al. (2019) [27] is a six primer multiplex PCR, which incorporates an internal snail control and offers a xenomonitoring tool for *S. haematobium* group species that are transmitted by *Bulinus* snail hosts. Our study simplifies the multiplex process, reducing the primer numbers and mitigating against PCR competition and some of the biases that may occur with multiple primer combinations. It also allows for greater amplicon size differentiation (as amplicon sizes can be more easily distinguished based on size), making results more interpretable. The ITS rDNA is a favorable target within the repeat ribosomal operon of *Bulinus* and *Schistosoma* spp., easily detected within small quantities of DNA due to the high copy number of rRNA clusters within eukaryote genomes. Another key feature of the target relates to specificity. The ITS regions of *Schistosoma* and *Bulinus* spp. can be routinely amplified using conventional PCR, thanks to its small size (~1000 bp) and highly conserved flanking regions (5′18S and 3′28S), enabling the use of universal primers (ETTS1 + 2) for multiple species [31]. However, interspecies heterogeneity and, to a lesser extent, intraspecies heterogeneity (Pennance et al., unpublished observations) of the ITS regions allow for differentiation between species, such as those of the *S. haematobium* group [7,33]. The internal *Schistosoma-*specific primer is situated in a conserved ITS region within the *Schistosoma* genus, with 100% conservation between several African species, suggesting that it could be utilized for several *Schistosoma-*snail transmission systems.

### 3.3. Limitations of Molecular Xenomonitoring Approaches for Schistosomiasis Surveillance

From our study, we have identified two limiting factors for the practical use of this method. First, the laborious nature of testing each individual snail adds time and cost. Further sensitivity testing should be performed to support the development of pooling strategies. This would help to determine whether infections are still detected when the *Schistosoma* DNA is diluted in the presence of much higher concentrations of snail DNA, which may inhibit the reaction. Pooling strategies have been successful for arthropod xenomonitoring protocols [18] and would allow for higher throughput of samples required for screening large populations of snails, such as those encountered for schistosomiasis.

Second, a limitation does come with the need for the secondary screening (SIX PCR) of the *Schistosoma* amplicon, via sequencing, to confirm species. Despite best efforts, rapid species diagnostics, such as the rapid diagnostic *cox*1 RD-PCR developed by Webster et al. (2010) [33] to determine adult worm and larval stage species identity, is not robust when snail DNA is present, particularly for *S. haematobium* infections. The *cox*1 RD-PCR was suggested as a secondary screening method by Schols et al. (2019) [27], but it was only theoretically examined as part of that study. Clearly, further ‘wet lab’ testing on infected snails is needed. In regions where *Schistosoma* hybridization occurs, mitochondrial DNA analysis would be necessary since both nuclear and mitochondrial DNA are required for hybrid identification [33]. Unfortunately, as with most diagnostics, there is a balance between sensitivity and specificity, with sensitivity increasing and specificity decreasing, usually due to the nature of the biomedical targets. Here, rapid screening with high sensitivity is a priority due to the low levels of infections in our study sites, with secondary species-specific screening only required on a subset of samples that are identified as infected. Moreover, Zanzibar previously thought to be a *S. haematobium* transmission only zone, although, with the recent report of *S. bovis* transmission [6], the additional species-specific screening is warranted. However, the need for the secondary screening step for *Schistosoma* species identification does need further exploration, such as trialing more direct methods that negate DNA sequencing, for example, amplicon enzyme restriction digestion demonstrated in Kane et al. (2013) [54]. However, it is also important to gather detailed information, as is obtained through DNA sequencing and analysis [6,7]. It is likely that xenomonitoring methods may need to be adapted to optimize focal surveillance strategies to specific endemic zones due to geographical genetic differences of the target organisms and potential unidentified species.

## 4. Materials and Methods

### 4.1. Primer Selection and In Silico Evaluation

The universal primer pair, ETTS2 and ETTS1 (Table 2 and Figure 7), was selected for the development of the internal control for the assay. They anneal to conserved flanking regions either side of the ITS(1 + 2) rDNA region of *Schistosoma* spp., amplifying the full ITS rDNA regions, resulting in an amplicon of ~1005 bp [6,7,31,33]. These primers have also been demonstrated to amplify the full ITS rDNA region of other organisms, including intermediate gastropod hosts. Primer’s cross-reactivity with the target *Bulinus* snail hosts was further confirmed through alignments of the ETTS2 and ETTS1 primers with *B. globosus* and *B. nasutus* rDNA regions, available from ongoing projects (Briscoe et al. unpublished data, Pennance et al. unpublished data). 

To develop the *Schistosoma-*specific target, two *Schistosoma* specific primers (ITS2_Schisto_F and ITS2_Schisto_R), published by Schols et al. (2019) [27], were selected, targeting the internal ITS1 and ITS2 rDNA regions of *Schistosoma* (Figure 7). These were further tested in silico for specificity by stringently aligning them with rDNA sequence data (Briscoe et al., unpublished data; Pennance et al., unpublished data) of a single *B. globosus* and *B. nasutus* from both Unguja and Pemba island (Zanzibar, United Republic of Tanzania) and those previously published for *Schistosoma* spp. [59,60].

All alignments were performed using Sequencher v5.4.6 (Gene Codes Corporation, Michigan, USA), and the primer positions were used to predict the specific amplicon sizes that would result, following amplification of snail and schistosome DNA using the different primer combinations of ETTS1, ETTS2, ITS2_Schisto_F, and ITS2_Schisto_R.

### 4.2. Bulinus and Schistosoma Genomic DNA Extractions

Whole soft tissue from *Bulinus* samples (as detailed below) available through the Schistosomiasis Collection at the Natural History Museum (SCAN) [61] and other ongoing projects, including laboratory and field samples, infected/non-infected and patent/non-patent, were used for the assay development and validation. Genomic DNA (gDNA) from all *Bulinus* samples was extracted using a modified tissue lysis protocol [6]. Two kits were then used to extract total gDNA from the lysed snail tissue—the BioSprint 96 DNA Blood Kit (Qiagen, Manchester, UK) for high-throughput multiple sample processing, and the DNeasy Blood & Tissue Kit (Qiagen, Manchester, UK) for single sample processing. Protocols were carried out according to the manufacturer’s instructions.

Positive control *Schistosoma* gDNA was obtained from adult worms; *S. haematobium* (single female worm from Zanzibar) and *S. bovis* (single male worm from Senegal) were available from SCAN. DNA was extracted following the DNeasy Blood & Tissue Kit protocol according to manufacturer’s instructions (Qiagen, Manchester, UK) [60].

### 4.3. PCR Conditions, Amplicon Visualization, and Sequencing

All PCR amplifications were performed in 25 µL reactions using illustra^TM^ PuReTaq Ready-To-Go^TM^ PCR Beads (GE Healthcare, UK) with 1 µL of each primer, in their different combinations, as stated in each section, at a concentration of 10 µM. gDNA templates (*Schistosoma* and/or *Bulinus* sp.) were added at different volumes and concentrations, as detailed below. The PCR cycling conditions for all multiplex and singleplex reactions were as follows: initial denaturation 5 min at 95 °C, followed by 40 cycles of 30 s at 95 °C, 30 s at 58 °C (unless stated otherwise), 90 s at 72 °C, and a final extension of 10 min at 72 °C. The visualization of all PCR products was performed by running 7.5 µL of each PCR product, mixed with 2 µL of Bioline 5x DNA Loading Buffer Blue (London, UK) and GelRed for visualization under UV light, on a 2% agarose gel for 90 min at 90 V. HyperLadder I and HyperLadder IV were run alongside the PCR amplicons to assess fragment sizes. Gels were visualized using a GBOX-Chemi-XRQ gel documentation system (Syngene, Cambridge, UK).

To validate amplification specificity, selected PCR amplicons from multiplex PCRs, where multiple amplicons were present, were cut from agarose gels and sequenced following purification using the QiaQuick Gel Purification Kit (Qiagen, Manchester, UK) following manufacturer’s instructions. For singleplex reactions, resulting in a single amplicon, PCR products were purified using the QiaQuick PCR Purification Kit (Qiagen, Manchester, UK) following the manufacturer’s instructions. The amplicons were sequenced in both directions using dilutions of the PCR primers. The sequence data were manually edited using Sequencher v5.4.6 (Gene Codes Corporation, Michigan, USA), and the amplicon identification was confirmed by comparison to *Schistosoma* reference data [59] and by BLAST analysis (BLAST: Basic Local Alignment Search Tool, NCBI).

### 4.4. In Vitro Primer Testing and Assay Validation

All gDNA extractions from laboratory-bred *Bulinus wrighti* (not exposed to any trematodes and, therefore, negative for infection) and from the *S. haematobium* and *S. bovis* adult worms were quantified using a Qubit^®^ Fluorometer using the dsDNA Broad Range (BR) Assay Kit (Molecular Probes, Life Technologies). The gDNA extracts from the single adult *S. haematobium* and *S. bovis* worms were normalized, using nuclease-free water, to 2 ng/µL (± 0.05 ng/µL). The gDNA extract of a *B. wrighti* snail control was recorded and kept at 31.3 ng/µL. Template gDNA (1 µL) was used in each PCR separately or combined and used to test the different primer combinations (shown in Figure 1). The primers were tested as singleplex PCRs for the internal control (ETTS2 + ETTS1), targeting both snail and *Schistosoma* gDNA, and then as multiplex PCR’s incorporating each of the internal *Schistosoma-*specific primers (ETTS2 + ETTS1 + ITS2_Schisto_F or ITS2_Schisto_R). All test PCRs were initially performed at an annealing temperature of 55 °C.

The multiplex primer combination ETTS2 + ITS2_Schisto_F + ETTS1 was selected and taken forward for further development and validation. This is referred to as the multiplex ITS xenomonitoring (MIX) PCR. The MIX PCR was further tested at annealing temperatures of 58 °C and 60 °C to enhance assay specificity, with 58 °C taken forward for further experiments. Additionally, a secondary *Schistosoma* ITS xenomonitoring (SIX) PCR, incorporating just the *Schistosoma-*specific primer (ITS2_Schisto_F) and its universal reverse primer (ETTS1), was validated, targeting just the 538 bp *Schistosoma* DNA amplicon. The SIX PCR was developed to obtain more targeted schistosome species data amplicon sequence analysis, of positive samples, following an initial high-throughput screening of snail populations with the multiplex PCR, which incorporates the internal snail control.

### 4.5. Sensitivity Testing

The analytical sensitivity and LoD for *Schistosoma* DNA in the MIX PCR were performed using serial dilutions of *S. haematobium* and *S. bovis* gDNA. The *S. haematobium* and *S. bovis* gDNA, normalized to 2 ng/µL (± 0.05 ng/µL), was diluted using nuclease-free water by one in ten (0.2 ng/µL), one in one hundred (0.02 ng/µL), and one in one thousand (0.002 ng/µL). 1 µL of each *Schistosoma* gDNA dilution was used in each multiplex PCR together with 1 µL of the *B. wrighti* gDNA (31.3 ng/µL).

Sensitivity was tested using controlled laboratory snail infections. Infections were performed by the Schistosomiasis Resource Center (SRC) (Biomedical Research Institute, Maryland, USA [62]) using their *B. truncatus*/*S. haematobium* (Egyptian strain) model lifecycle system. Juvenile *B. truncatus* (2–3 mm, *n* = 133) was divided into three groups, with individual snails in each group being exposed to either 1, 2, or several (~7) *S. haematobium* miracidia, respectively (Table 3). Miracidia, hatched in freshwater from eggs collected from *S. haematobium-*infected male LVG Syrian golden hamsters (see Ethical Statement), were added to individual wells of 24-well ELISA plates containing the *B. truncatus* snails. A fine-tipped Pasteur pipette was used under a dissection microscope to capture and deliver either an individual miracidium (for 1 and 2 miracidia exposures) or several (~7) miracidia at a time, following the standard operating procedures (SOPs) conducted at SRC (see: https://www.afbr-bri.org/schistosomiasis/standard-operating-procedures/).

The snails were kept in their individual wells until no miracidia were observed swimming under a binocular microscope, assumed to have penetrated the snail (~2 h). Following 24 h after initial exposure to the miracidia, half of each infection group was preserved in 100% ethanol for molecular analysis. The remaining exposed *B. truncatus* were maintained in their separate infection groups for 11 weeks to allow the infections to mature, and since this was the first opportunity to conduct sampling of the infected snails. Snails were maintained according to the SRC’s SOP’s (see above). Snails that died were recorded and promptly removed from the group. At 11 weeks post-exposure, the remaining snails were individually induced to shed cercariae by exposure to freshwater and light. Once it had been determined if the snails were infected and patent, they were washed, to remove any cercariae, and preserved in 100% ethanol for molecular analysis.

The MIX PCR was performed using gDNA (1 µL) extracted from six individual *B. truncatus* from each group, which were preserved after 24 h—two non-patent snails from group 1 and 2, and one non-patent snail from group 3 (11 weeks post-exposure), and one patent (shedding) snail from group 2 (11 weeks post-exposure) (Table 3). The secondary SIX PCR was performed on selected *Schistosoma* positive samples, to amplify the 538 bp *S. haematobium-*specific amplicon for sequence analysis to confirm that the MIX PCR was not a false-positive.

### 4.6. Specificity Testing and Validation on Field Samples

As part of a longitudinal xenomonitoring project on Pemba in relation to urogenital schistosomiasis transmission [6], the ‘wild-caught’ *B. globosus* and *B. nasutus* field isolates were available for further validation of the MIX assay. Individual snails had been collected during malacological surveys, individually checked for patent trematode infections by cercarial shedding, and then preserved in 100% ethanol for molecular analysis [6]. Cercariae from infected *B. globosus* were preserved on Whatman FTA cards and identified using molecular methods as *S. haematobium* or *S. bovis* from two and five snails, respectively [6]. In addition, individual *B. globosus* and *B. nasutus* (also collected from Pemba), which were shedding two other trematode species, *Euclinostomum* sp. and *Echinostoma* sp., respectively (unpublished data), were tested to investigate assay specificity. Additionally, 94 *B. globosus* snails from Wambaa (Pemba) collected during November 2018, which were not shedding any trematode cercariae, were tested for infections by PCR. 

All samples, which gave the 538 bp *Schistosoma-*specific amplicon (Figure 7), were further subjected to the SIX PCR assay with the resulting amplicons purified and sequenced to confirm the species of the infection. The identity of the *S. haematobium* and *S. bovis* species was confirmed by analysis of the four species’ SNPs that exist in the ITS2 region [7] between *S. haematobium* and *S. bovis* (Table 1).

### 4.7. Testing the Schistosoma cox1 Rapid-Diagnostic PCR (RD-PCR) for Secondary Species Identification

The patent *B. globosus* snails collected from Pemba shedding either *S. haematobium* (*n* = 2) or *S. bovis* (*n* = 5) (see [6]), as detailed above, were further tested using the published multiplex RD-PCR (see [27,33]) with an aim to provide a secondary species-specific screening method, as described in Schols et al. (2019) [27]. This multiplex RD-PCR, capable of differentiating *S. bovis* and *S. haematobium* by species-specific amplicon size (*S. haematobium* (543 bp), *S. bovis* (306 bp)), was performed following the published protocol and cycling conditions described by Webster et al. (2010) [33]. Different amount of gDNA (1 µL, 2 µL, and 3 µL) and PCR annealing temperatures (58 °C, 62 °C, and 65 °C) were trialed to investigate sensitivity and specificity. The amplicons were purified, and Sanger sequenced, as described above, using the species-specific reverse primers to confirm species/amplicon identification.

### 4.8. Ethical Statement

*Schistosoma haematobium* experimental infections were conducted at the Biomedical Research Institute – Schistosomiasis Resource Center (Rockville, MA, USA) animal facility maintained with AAALAC full accreditation (Site # 000779), operating under the National Institutes of Health’s Office of Laboratory Animal Welfare (OLAW) # A3080-01. *S. haematobium* parasite material was collected from male LVG Syrian golden hamsters following percutaneous exposure to cercariae. Hamster’s use was approved by the Institutional Animal Care and Use Committee (IACUC) of the Biomedical Research Institute for the Animal Use Protocol, #18-01.

## Figures and Tables

**Figure 1 molecules-25-04011-f001:**
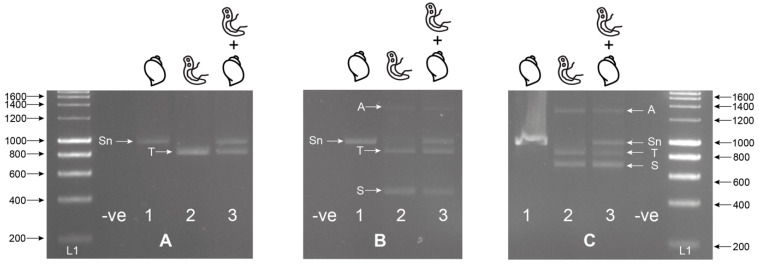
Singleplex (**A**); ETTS2 + ETTS1) and multiplex (**B**); multiplex ETTS2 + ETTS1 + ITS2_Schisto_F, (**C**); ETTS2 + ETTS1 + ITS2_Schisto_R) PCRs on laboratory-bred *Bulinus wrighti* (*B.w.*) and *Schistosoma haematobium* (*S.h.*) gDNA separately (1; *B.w.*, 2; *S.h.*) and combined (3; *B.w.* + *S.h.*). When *B.w.* and *S.h.* DNA was combined (A3, B3, C3), two amplicons were produced by the ETTS1 + ETTS2 primers, a larger snail amplicon (Sn) (~1200 bp) and a smaller *Schistosoma* amplicon (T) (~1000), with the additional *Schistosoma-*specific primers producing either a 538 bp (B3; ITS2_Schisto_F) or 835 bp (C3; ITS2_Schisto_R) amplicon (S). A larger amplicon (A) (~1400–1600 bp) was also observed to be amplified in some reactions, and this was thought to be a PCR artifact or additional primer targets in the *Schistosoma* gDNA. L1 = HyperLadder I (Bioline, London, UK). -ve = negative, no template control. ITS = internal transcribed spacer.

**Figure 2 molecules-25-04011-f002:**
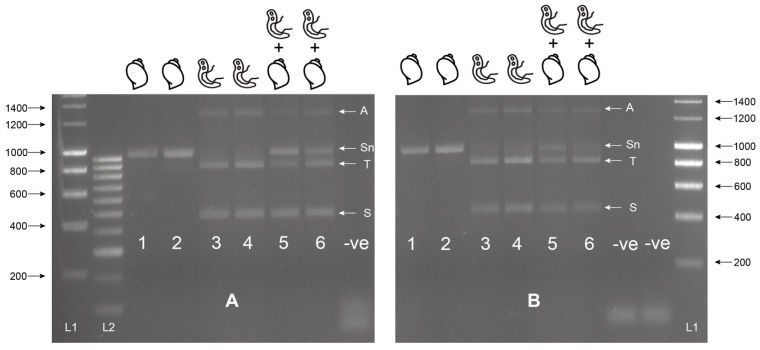
Multiplex ITS xenomonitoring assay trial at 55 °C (**A**) and 60 °C (**B**). Includes gDNA of *Bulinus wrighti* of both BioSprint (Lane 1 and 5) and DNeasy extractions (Lane 2 and 6) and gDNA of *Schistosoma haematobium* (Lane 3 and 5) and *S. bovis* (Lane 4 and 6). Combinations of *B. wrighti* and *S. haematobium* (Lane 5) or *S. bovis* (Lane 6) gDNA shown. Sn = snail amplicon, T = trematode amplicon, S = *Schistosoma* amplicon, and A = non-specific amplicon or artifact. L1 = HyperLadder I. L2 = HyperLadder IV (Bioline, London, UK). -ve = negative, no template control.

**Figure 3 molecules-25-04011-f003:**
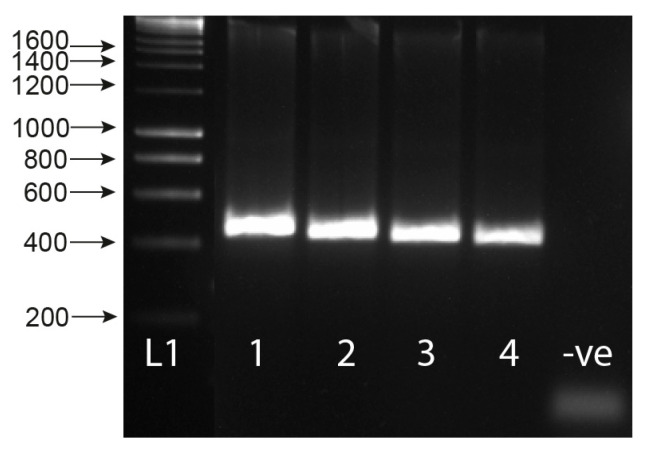
Gel showing the secondary singleplex ITS xenomonitoring (SIX) PCR for 1) *Schistosoma haematobium* gDNA; 2) *S. bovis* gDNA; 3) *S. haematobium* + *B. wrighti* gDNA; 4) *S. bovis* + *B. wrighti* gDNA. -ve = non-template negative control. L1 = HyperLadder I (Bioline, London, UK).

**Figure 4 molecules-25-04011-f004:**
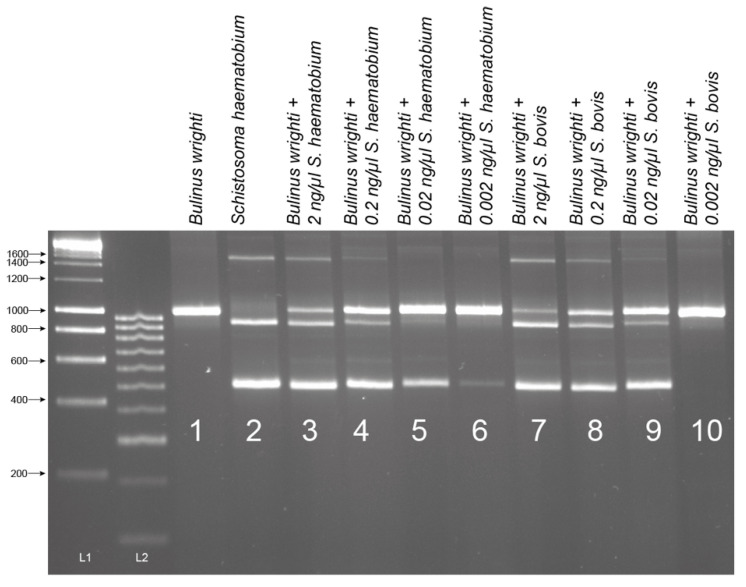
Sensitivity tests of ITS1-2-F PCR performed with serial dilutions of *Schistosoma haematobium* and *S. bovis* gDNA in the presence of *Bulinus wrighti* gDNA. L1 = HyperLadder I. L2 = HyperLadder IV (Bioline, London, UK).

**Figure 5 molecules-25-04011-f005:**
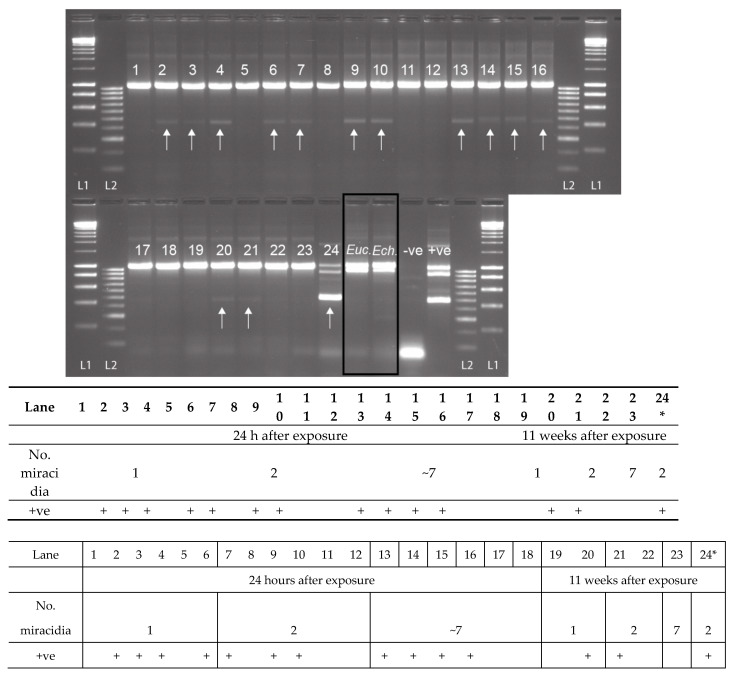
Experimental infections of *Bulinus truncatus* with *Schistosoma haematobium* (1–24), field-collected *B. globosus* infected with *Euclinostomum* sp. (*Euc*.) and field-collected *B. nasutus* shedding *Echinostoma* sp. cercariae (*Ech*.). The *S. haematobium* DNA amplicon was present (+ve) in 13 of the 23 non-patent snails (11 at 24 h post-exposure, and two at 11 weeks post-exposure), highlighted by the arrow. Lane 24 = *B. truncatus* sample that was shedding *S. haematobium* cercariae 11 weeks after exposure. The positive control (+ve) is a mix of *B. wrighti* and *S. haematobium* control gDNA. L1: HyperLadder I, L2: HyperLadder IV (Bioline, London, UK).

**Figure 6 molecules-25-04011-f006:**
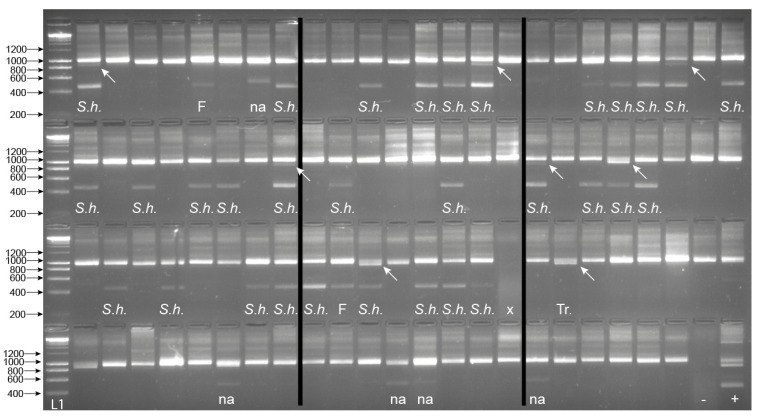
Gel images for the multiplex ITS xenomonitoring (MIX) PCR amplicons for 94 non-patent *Bulinus globosus* collected from Wambaa, Pemba, United Republic of Tanzania. The text under each amplicon denotes the outcome of the *Schistosoma* sp. targeted sequencing where relevant (i.e., presence of 538 bp amplicon), which resulted in either *S. haematobium* (*S.h*.) or sequencing failure (F). The presence of a trematode band without the presence of the *Schistosoma* band indicated a non-*Schistosoma* trematode infection (Tr.). Other non-specific bands, in this case, larger bands (NA), were also observed in these snail populations, which did not amplify with the secondary SIX PCR. x = sample failure with no control amplicon. Arrows highlight the presence of the ~1000 bp trematode band when present (*n* = 8). *B. globosus* with a patent *S. haematobium* infection (Cham10.1 see [6]) was run as a positive control (+ve) and also represented the amplicon profile obtained for the seven patent *B. globosus* snails (five and two with *S. bovis* and *S. haematobium* infections, respectively (see Section 2.4). -ve = the non-template negative control. L1—HyperLadder IV (Bioline, London, UK).

**Figure 7 molecules-25-04011-f007:**
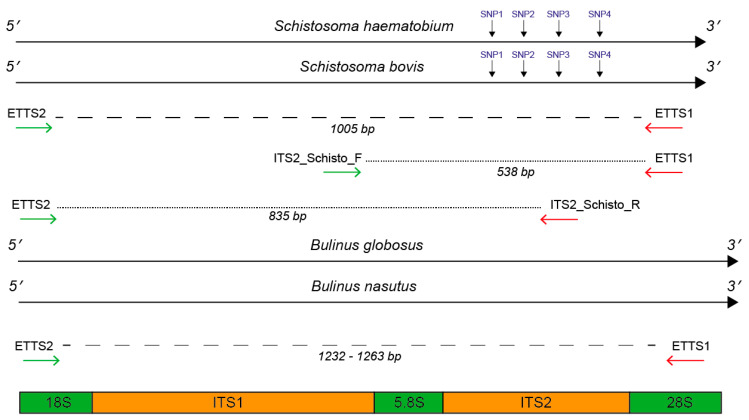
Primer annealing positions flanking and internal to the ITS1 + 2 rDNA targets. Primer positions were mapped to *Schistosoma haematobium* and *S. bovis* ITS1 + 2 reference data [59] and to a *Bulinus globosus* and *B. nasutus* DNA reference (Pennance et al., unpublished data). For *Schistosoma* DNA, the primer combinations produced two fragments; 1) ETTS2–ETTS1 (1005 bp) and either 2) ITS2_Schisto_F-ETTS1 (538 bp) or 3) ITS2_Schisto_R-ETTS2 (835 bp). For *Bulinus* DNA, the primer combinations produced one fragment, ranging in size between 1232 and 1263 due to interspecies variation. For *Schistosoma* species identification, four SNPs were present at bp positions 90, 145, 195, and 265 in the ITS2 rDNA region, allowing differentiation of *S. haematobium* and *S. bovis* following ITS2 sequencing.

**Table 1 molecules-25-04011-t001:** *Schistosoma* species-specific SNP positions (including base position) in the internal transcribed spacer (ITS)2 region.

Schistosoma Species	ITS 2 Schistosome Species-Specific SNP Positions (bp)			
	SNP1 (90)	SNP2 (145)	SNP3 (195)	SNP4 (265)
S. haematobium	S. h (G)	S. h (C)	S. h (G)	S. h (C)
S. bovis	S. b (A)	S. b (T)	S. b (A)	S. b (T)

**Table 2 molecules-25-04011-t002:** Details of the primers selected for the development of the xenomonitoring assay. Universal (U) and specific (S) denote whether the primers universally target both *Schistosoma* and snail or just specifically target *Schistosoma* DNA.

Primer (Direction)	Primer Sequence (5′-3′)	Primer Position	State	Reference
ETTS1 (Reverse)	TGCTTAAGTTCAGCGGG	28S 5′ end (ITS2 3′ flanking region)	U	Kane et al. (1994) [31]
ETTS2 (Forward)	TAACAAGGTTTCCGTAGGTGA	18S 3′ region (ITS1 5′ flanking region)	U	Kane et al. (1994) [31]
ITS2_Schisto_F (Forward)	GGAAACCAATGTATGGGATTATTG	ITS1 3′ end (5.8S 5′ flanking region)	S	Schols et al. (2019) [27]
ITS2_Schisto_R (Reverse)	ATTAAGCCACGACTCGAGCA	ITS2 (middle)	S	Schols et al. (2019) [27]

**Table 3 molecules-25-04011-t003:** Groups of *Bulinus truncatus* (*B.t.*) experimentally challenged with either 1, 2, or ~7 *S. haematobium* (*S.h*.) miracidia and preserved 24 h post-exposure or checked for patent *S.h.* infections and preserved 11 weeks (wks) post-exposure.

Infection Group	No. of *B.t.* Exposed	No. of *S.h.* Miracidia Used	No. of *B.t.* Preserved at 24 h	No. of *B.t.* Checked for Patency at 11 wks and Preserved (no. Shedding + ve)
1	45	1	22	22 ^1^ (0)
2	43	2	21	19 ^1^ (2)
3	45	~7	23	21 ^1^ (7)

^1^ One *B. truncatus* died from each infection group during the 11 weeks post miracidia exposure.

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
