# Peer review of "Development of a Molecular Snail Xenomonitoring Assay to Detect Schistosoma haematobium and Schistosoma bovis Infections in their Bulinus Snail Hosts"

_molecules, 2020, doi:10.3390/molecules25174011_

Round 1

Reviewer 1 Report

The manuscript by Pennance and coauthors presents a refinement of current molecular detection methods for snail-infected schistosomes along with some additional controls not found in previous similar reports. Overall, the work seems sound, though there are some concerns, outlined below.

1. It was not clear to me why Table 3 was buried in the Methods section instead of being in the Results section. If some further key information is added to it (eg, number of SIX-positive/negative, etc.), it could serve as a nice summary of the results.

2. It would have been nice if there was someindependent parasitological confirmation that the PCR-positive field samples were actually infected with schistosomes (eg, by splitting tissues, using some for DNA extraction, the rest for microscopic examination). As it stands, the conclusions of accuracy seem somewhat circular.

3. Although the authors cite some papers using isothermal molecular amplification (eg, LAMP), there is no discussion of the major logistical and sensitivity advantages of those approaches, particularly in a point-of-contact setting. It might also be helpful to mention possible adaptation of newer, point-of-care qPCR technology that has become available (eg, Biomeme). As it stands, the approach described in this report seems to be quite labor intensive, requiring multiple steps and trained personnel.

Minor points:

line 33 - ITS should be spelled out in the abstract

line 46 - change "genus" to "genera"

line 49 - insert comma after "schistosomiasis"

line 60 - praziquantel should not be capitalized

line 61 - insert "asexual" after "prolific"

line 94 - vitro should not be capitalized

line 124 - delete "and"

line 150 - change "assays" to "assay's"; change "miracidia" to "miracidium"

line 151 - delete "It was also observed that'; change "at" to "At"; insert "DNA" after "Schistosoma"

line 161 - delete ", but"

line 162 - insert "but" before "were"

line 164 - delete comma after "infected"

line 182 - is "results not shown" acceptable for this journal?

line 245 - change "smaller" to "lower"

line 250 - delete commas after "tested" and after "infections"

line 296 - change "homology to" to "conservation between"

line 433 - insert "an" after "either"

Author Response

Reviewer 1: Responses in bold

The manuscript by Pennance and coauthors presents a refinement of current molecular detection methods for snail-infected schistosomes along with some additional controls not found in previous similar reports. Overall, the work seems sound, though there are some concerns, outlined below.

We would like to thank reviewer one for their in depth line-by-line corrections and also general concerns. We appreciate the time they have spent in overlooking the manuscript. We hope the responses to their comments and changes made to the manuscript are to their satisfaction.

  1. It was not clear to me why Table 3 was buried in the Methods section instead of being in the Results section. If some further key information is added to it (eg, number of SIX-positive/negative, etc.), it could serve as a nice summary of the results.

We can appreciate where reviewer 1 is coming from here, however the reason table 3 is in the methods is because the experimental infections were used as an additional proof of concept for the xenomonitoring assay rather than as a primary output. The number of snails that were challenged and then developed patent infections was part of the methodology to create natural experimentally infected Bulinus. Also, not all the challenged snails were used in this study, and only a subset trialled, therefore adding the number of SIX positives from the subset is a result presented in the results section 2.3.

  1. It would have been nice if there was some independent parasitological confirmation that the PCR-positive field samples were actually infected with schistosomes (eg, by splitting tissues, using some for DNA extraction, the rest for microscopic examination). As it stands, the conclusions of accuracy seem somewhat circular.

We fully agree that this would have been a nice additional proof of concept for the xenomonitoring marker study. However, this process is even more labour intensive, unfortunately making this not possible in the current study since we were conducting high numbers of whole snail tissue extractions. For accuracy, this would also require extensive laboratory time for a skilled parasitologist with experience in conducting these extractions before, which was not possible here. Additionally, the dissections may have created issues in then performing the whole snail tissue extractions required for accurate schistosome DNA detection. Both have their advantages and disadvantages.

  1. Although the authors cite some papers using isothermal molecular amplification (eg, LAMP), there is no discussion of the major logistical and sensitivity advantages of those approaches, particularly in a point-of-contact setting. It might also be helpful to mention possible adaptation of newer, point-of-care qPCR technology that has become available (eg, Biomeme). As it stands, the approach described in this report seems to be quite labor intensive, requiring multiple steps and trained personnel.

We thank the reviewer for drawing our attention to the work being conducted using Biomeme technology. We have added reference to the use of these methods in the research conducted by Rudko et al., and have added a sentence to the discussion in section 3.2 to highlight this:

“However, recent technological advances in sample preparation and DNA extraction methods have demonstrated robust field setting methodologies to conduct downstream qPCR analysis capable of detecting avian trematodes and host species in Canadian lakes [56–58], which could potentially be modified to suit the detection of human and bovine schistosomes in sub-Saharan Africa, although cost and throughput would need to be considered.”

We do believe however that both PCR and qPCR methods have their advantages and disadvantages for use in endemic settings, due to accessibility and cost of differences of both methods. These are presented fairly in the discussion. We also want to draw attention to the reviewer that the methodology presented here is geared towards a high-throughput method for the extraction (Qiagen Biosprint) and PCR of whole plates of samples (~96 samples), which does not seem possible using Biomeme technology.

Minor points:

line 33 - ITS should be spelled out in the abstract. DONE

line 46 - change "genus" to "genera". Genus is the correct term used in this instance as referring to the single genus Schistosoma, genera is plural

line 49 - insert comma after "schistosomiasis" DONE

line 60 - praziquantel should not be capitalized DONE

line 61 - insert "asexual" after "prolific" DONE

line 94 - vitro should not be capitalized DONE – this was changed by the editing team at Molecules journal after submission

line 124 - delete "and" DONE.

line 150 - change "assays" to "assay's"; change "miracidia" to "miracidium" DONE

line 151 - delete "It was also observed that'; change "at" to "At"; insert "DNA" after "Schistosoma" DONE

line 161 - delete ", but" DONE

line 162 - insert "but" before "were" DONE

line 164 - delete comma after "infected" DONE

line 182 - is "results not shown" acceptable for this journal? Using results not shown seems appropriate here if the journal will accept for two reasons, it avoids the need for another gel image and also in the context of this xenomonitoring marker, it is irrelevant to show a PCR positive schistosome infection in a snail, when having visual confirmation of a snail shedding cercariae.

line 245 - change "smaller" to "lower" DONE

line 250 - delete commas after "tested" and after "infections" DONE

line 296 - change "homology to" to "conservation between" DONE

line 433 - insert "an" after "either" DONE

Reviewer 2 Report

Considering current situation of schistosomiasis in endemic foci, monitoring with molecular tools are expected.  The aim of the study is to develop an easy and reliable molecular monitoring method for Schistosoma haematobium group.  Points to be uncover for their purpose are clear and results shown here are informative for countermeasure of the disease.

Several small points seem to be considered by the authors.

(1) Biologically, schistosome-host snail interaction is rather critical, therefore, the geographical origin of parasites and snails used in the study should be clarified.

(2) As mentioned by authors, species identification is still remained to be achieved.  Is it possible to show the actual result figure for the testing?

(3) Related to the above comment, recent report of the hybrid species of S. haematobium and S. bovis, species identification is becoming more important, although it is not their current purpose. 

Author Response

Reviewer 2: Responses in bold

Considering current situation of schistosomiasis in endemic foci, monitoring with molecular tools are expected.  The aim of the study is to develop an easy and reliable molecular monitoring method for Schistosoma haematobium group.  Points to be uncover for their purpose are clear and results shown here are informative for countermeasure of the disease.

We thank reviewer 2 for taking the time to read over and offer suggestions for the manuscripts improvement.

Several small points seem to be considered by the authors.

(1) Biologically, schistosome-host snail interaction is rather critical, therefore, the geographical origin of parasites and snails used in the study should be clarified.

The geographical origin of the snails and parasites is indeed crucial, these are clearly outlined in the methods section for each experimental infection, template DNA used and field collected snails.

(2) As mentioned by authors, species identification is still remained to be achieved.  Is it possible to show the actual result figure for the testing?

We presume reviewer 2 is referring to the species identification of the schistosome rather than the snail. Schistosome species identification through RD- PCR proved difficult, as outlined in section 2.6 and later in the discussion when discussing propositions by Schols et al. for using this marker in species identification. As very few samples successfully amplified here using this methodology, it is was not deemed necessary to show the gel images, especially since these gel images are displayed in previous publications by Webster et al. (2010) Journal of Helminthology, for which this references.

(3) Related to the above comment, recent report of the hybrid species of S. haematobium and S. bovis, species identification is becoming more important, although it is not their current purpose. 

Thanks to reviewer 2 for this valid point. The RD-PCR of the cox1 of the schistosomes would not be able to differentiate between pure and hybrid species, as this would require sequencing of a nuclear DNA region such as the ITS1-2 as well as a mitochondrial gene with species specific SNPs. As outlined in the results, we have demonstrated the capacity for nuclear DNA species identification by amplifying and sequencing the ITS2 region in the SIX PCR. Therefore, a mitochondrial gene identification is still warranted when using this xenomonitoring marker on snail extracts in endemic settings where hybrid schistosomes are present (e.g. West Africa). We have added a sentence to section 3.3 to highlight this:

“In regions where Schistosoma hybridisation occurs, mitochondrial DNA analysis would be necessary, since both nuclear and mitochondrial DNA is required for hybrid identification [33].”

Reviewer 3 Report

well constructed research approach and well written- minor typos e.g line 245 space between 0.02 and ng

Author Response

Reviewer 3: Response is in bold

well constructed research approach and well written- minor typos e.g line 245 space between 0.02 and ng

Thanks to reviewer 3 for taking the time to read and review the paper. We have corrected the typo on line 245.